# The 25(OH)Vitamin D Status Affected the Effectiveness of Oligo Fucoidan in Patients with Chronic Hepatitis B Virus Infection with Immune Tolerance Phase

**DOI:** 10.3390/nu12020321

**Published:** 2020-01-26

**Authors:** Wang-Sheng Ko, Fang-Ping Shen, Chia-Ju Shih, Ya-Ling Chiou

**Affiliations:** 1Department of Nutrition, Master of Biomedical Nutrition Program, HungKuang University, Taichung 43302, Taiwan; ker200448@yahoo.com.tw (W.-S.K.); a0939631309@gmail.com (F.-P.S.); sgazo063@hotmail.com (C.-J.S.); 2Department of Internal Medicine, Kuang-Tien General Hospital, Taichung 43302, Taiwan

**Keywords:** chronic hepatitis B, oligo fucoidan, vitamin D, T lymphocytes

## Abstract

Chronic hepatitis B virus (HBV) infection is a serious public health issue. Vitamin D is involved in various pathophysiological mechanisms as an immune modulator and the deficiency rate of vitamin D is prevalent in chronic liver disease. Fucoidan exerts anti-inflammatory, anticoagulant, antitumor, antimetastatic, and antiangiogenetic effects; however, its effect on the immune responses of HBV patients is unclear. This study investigated how 25(OH)Vitamin D status affected the effectiveness of oligo fucoidan in patients with HBV infection in the immune tolerance phase. Fifty-one patients received oligo fucoidan 4400 mg/day for 48 weeks. Flow cytometry was used to detect T lymphocyte markers (CD3^+^CD4^+^, CD3^+^CD8^+^, CD4^+^CD45RO^+^, CD8^+^CD45RO^+^). The levels of white blood cell (WBC), platelets (PLT), and albumin were decreased after 48 weeks of supplementation (*p <* 0.05). Percentages of CD3^+^CD8^+^ and CD8^+^CD45RO+ cells were decreased after 12 weeks of supplementation (*p <* 0.05). In patients with adequate vitamin D, HBV-DNA concentrations decreased and the proportion of CD4^+^CD45RO^+^ and CD8^+^CD45RO^+^ cells increased upon oligo fucoidan supplementation. The HBeAg status of one vitamin D-adequate patient changed from positive to negative at the 12th week of supplementation. The oligo fucoidan may regulate immune effects in patients with HBV infection, and the 25(OH)Vitamin D status might have affected the effectiveness of oligo fucoidan.

## 1. Introduction

Hepatitis B virus (HBV) infection is a global disease. Approximately two billion people worldwide have been infected with HBV, more than 350 million people are chronic carriers [1], and more than 780,000 die each year from acute or chronic HBV infection [2]. In North America, the prevalence of asymptomatic hepatitis B in adults is 0.5%; the prevalence of asymptomatic adults in Taiwan is about 15%, indicating that Taiwan is a high-prevalence area of HBV infection. A hepatitis B vaccine program for newborns with hepatitis B surface antigen-positive mothers has existed since July 1984 in Taiwan. In July 1986, hepatitis B vaccination of infants and young children was fully implemented. Thereafter, the hepatitis B carrier rate of six-year-old children significantly decreased from 10.5% in 1989 to 0.8% in 2007, but there are still 2.8 million to 3.3 million chronic hepatitis B carriers (CHB). Chronic hepatitis B carriers are a high-risk group for cirrhosis and hepatocellular carcinoma. About 50% of CHB die from liver cancer or cirrhosis in Taiwan [3].

Hepatitis B virus is a deoxyribonucleic acid (DNA) virus with a diameter of 42 nm. The virus is covered with a lipoprotein coat, which contains the hepatitis B surface antigen (HBsAg). Another hepatitis B-related antigen is the hepatitis B virus e antigen (HBeAg), a soluble protein that is produced during the replication of the hepatitis virus and is secreted into the blood. The primary purpose of treatment of hepatitis B is to chronically inhibit the blood virus, which can reduce the fibrosis caused by inflammation and even prevent liver cancer. Patients who are positive for viral HBV-DNA are treated with a condition in which liver function (alanine aminotransferase; ALT) is greater than twice (80 units/L) the typical healthy upper limit (40 units/L), and ALT abnormalities persist for more than one month. Currently, the most important treatment is interferon injection or oral antiviral drugs to directly or indirectly inhibit the blood virus [4]. 

HBV initiates innate and adaptive immune responses in the host. Liver cells infected with HBV can secrete interferon-γ, tumor necrosis factor-α, and interferon-α/β, and inhibit HBV replication in liver cells. Liver cells also activate natural killer cells and natural killer T cells (NKT cells) and then activate T cells (especially cytotoxic T cells) and B cells [5]. According to previous research, patients with CHB have decreased CD4^+^ T cell and increased CD8^+^ T cell counts, and exhibit a negative correlation with the amount of virus in the blood [6]. In addition, some studies have found that HBV also reduces the number of CD45RO^+^ T cells (memory T cells) [7,8]. CD45RA and CD45RO are used as markers for different stages of lymphoblast differentiation. CD45RA is expressed on the surface of naïve T cells and CD45RO is expressed on the surface of memory T cells; in activation, CD45RO^+^ T cells release a number of lymphokines to trigger or maintain the immune response. Studies indicate that administration of lamivudine in combination with a high concentration of hepatitis B virus immunoglobulin (HBIG) in patients with chronic hepatitis B increases the number of CD45RO^+^ T cells [9]. Changes in the number of CD45RO^+^ T cells can be used to assess the effect of drugs or nutrients on the immune response.

At present, it is believed that the main reason for the difficulty in curing chronic hepatitis B is the immunopathology of patients [10]. The immune levels of patients are closely related to nutritional status. Vitamin D can stimulate an antiviral immune response and vitamin D deficiency can increase viral replication [11]. Vitamin D deficiency is extremely prevalent in chronic liver disease [12].

Fucoidan is a sulfated polysaccharide derived from brown seaweed extracts. It is structurally similar to a heparin molecule; it consists of repeating units of disaccharides containing an alpha–1, 3-linked fucose and an alpha–1, 4-linked fucose, forming an alpha–1, 3-backbone with branches attached at C2 positions. [13,14]. Fucoidan is reported to possess antiviral, antioxidant, antimicrobial, anticoagulant, anticancer, antitumor, antiproliferative, and anti-inflammatory properties [15,16,17,18]. The effect of fucoidan in the liver has also been extensively studied. Studies have shown that fucoidan can reduce liver fibrosis and have anti-inflammatory effects in patients with nonalcoholic hepatitis [19,20,21,22]. Fucoidan has been shown to protect the liver from free radical damage in cell and animal experiments on alcoholic hepatitis [23]. In addition, fucoidan has anti-liver cancer effects [24] and reduces liver damage caused by acetaminophen [25]. The use of fucoidan treatment for viral hepatitis has shown no adverse effects and has good therapeutic effects in patients with chronic hepatitis C [26]. It is not clear whether it is effective in patients with chronic hepatitis B; thus, the purpose of this study is to determine the effects of supplementation with fucoidan in patients with chronic hepatitis B and liver function ALT less than twice the healthy upper limit (>40 units/L and <80 units/L). We evaluated immune status, viral load, e-antigen, surface antibodies, and inflammation indicators.

## 2. Materials and Methods 

### 2.1. Study Subjects 

The patients were enrolled from among the outpatients of Kuang-Tien General Hospital. The following were used as the inclusion criteria: chronic hepatitis B carriers, HBeAg-positive or -negative, and ALT less than two times that of the normal value in healthy individuals (>40 units/L and <80 units/L). The exclusion criteria diagnosed by a physician included cirrhosis, liver cancer, hepatitis C, diabetes, immune diseases, other malignancies, other chronic diseases, infection, pregnancy, or other systematic immune diseases. Fifty-one patients were included in this study. Subjects received four tablets of oligo fucoidan (550 mg) twice a day for 48 weeks (Hi-Q oligo-fucoidan^®^ was a gift from Hi-Q Marine Biotech International Ltd., Taipei, Taiwan). All subjects were fully aware of the purpose and nature of the study, which was approved by the Institutional Review Board (IRB) of Kuang-Tien General Hospital.

### 2.2. Analysis of Basic and Biochemical Data of Subjects

Blood was collected from the 51 subjects to measure biochemical values at week 0 (before the experiment), and the 4th, 12th, 24th, 48th weeks (in the experiment) and in the 4th and 12th weeks after the experiment. Heparinized blood was collected to evaluate liver and kidney function and genotype, and to quantify HBV-DNA concentration, HBsAg, and HBeAg. Serum HBV-DNA levels (copies/mL) were measured by polymerase chain reaction (PCR) using Cobas TaqMan HBV Test, v2.0 (Roche Molecular Systems, Pleasanton, CA, USA), according to the manufacturer’s instructions. Quantification of HBsAg was performed by an automated chemiluminescent micro-particle immunoassay (CMIA) using the Roche Cobas e602 analyzer with Elecsys HBsAg II Quant reagent kits (Roche Diagnostics, Santa Clara, CA, USA), according to the manufacturer’s instructions. Quantification of HBeAg was conducted by an automated chemiluminescent micro-particle immunoassay (CMIA) using the Roche Cobas e602 analyzer with Elecsys HBeAg Quant reagent kits (Roche Diagnostics, CA, USA), according to the manufacturer’s instructions. The remaining blood was segregated into serum and cells. Serum was stored as aliquots in liquid nitrogen until analysis and cells were analyzed to determine the T cell subtypes.

### 2.3. 25(OH)Vitamin D Concentration Measurement

The level of 25(OH)Vitamin D was measured by Chemiluminescence assay. The serum 25(OH)Vitamin D concentrations of all subjects were divided into <20 ng/mL (Vitamin D deficiency group), 20–30 ng/mL (Vitamin D indeficiency group), and >30 ng/mL (Vitamin D normal group).

### 2.4. Analysis of T Cell Subtypes 

The cells were incubated with fluorescein isothiocyanate (FITC)-conjugated anti-human CD3, anti-human CD25, or anti-human CD45RO for 20 min at 4 °C in the dark, followed by staining with phycoerythrin (PE)-conjugated anti-human CD4, anti-human CD8, or CD19 antibodies, for separating individual T cell subtypes. PE-conjugated mouse IgG1 antibodies were used as isotype controls. These antibodies were purchased from BD Company (Franklin Lakes, NJ, USA). 

### 2.5. Statistical Analysis

If the continuous variable was normally distributed, we used an unpaired-t test for analysis and the value was expressed as mean ± SD. The continuous variables are repeated measurement data, and repeated measures are used for analysis. For the categorical variable, a chi-squared test was performed. If the total number of cells with less than 5 expected times in each cell exceeded 20%, Fisher’s exact test was required. The values of the categorical variable are expressed as numbers and percentages. *p <* 0.05 indicates a significant difference. 

## 3. Results

### 3.1. Patient Characteristics

A total of 51 patients with chronic hepatitis B were enrolled in this study. These patients take any medicine for HBV infection. Of the patients, 62.7% were supplemented with nutritional supplements, of which 85% were supplemented with comprehensive vitamins and minerals. Their ALT was less than twice the typical healthy value (<80 IU/L), and the presence of HBsAg was sustained for more than six months. The outpatient follow-up occurred over one year. All patients were in the immune tolerance phase and could not be administered interferon or oral antiviral therapy. The basic information of the study objects is summarized in Table 1. The analysis showed that most of the subjects were male (54.9%), the average age was 44.86 ± 10.81 years, and the average BMI was 24.78 ± 4.38 (kg/m^2^). With respect to 25(OH)VitaminD, the average serum 25(OH)Vitamin D concentration in the patients was 20.6 ± 6.1 ng/mL, which is inadequate (Table 1). Higher levels of white blood cell (WBC), platelets (PLT), and albumin were recorded at the baseline than after 48 weeks of fucoidan administration (*p <* 0.05). Higher levels of AC sugar (before meals sugar) were recorded at the baseline than after four weeks of fucoidan administration (*p <* 0.05) (Table 2). 

### 3.2. Biochemical Values Before and After Oligo Fucoidan Supplementation

The biochemical values of the subjects before and after oligo fucoidan supplementation are shown in Table 2. During oligo fucoidan supplementation, the levels of WBC, albumin, and platelets decreased from baseline, but these data were in the normal range. AC sugar, triglyceride (TG), and cholesterol did not show an upward or downward trend during the supplementation. The results also showed that the supplementation of oligo fucoidan had no significant effect on liver function (ALT and AST) and renal function (creatinine) of the subjects, illustrating the safety of oligo fucoidan. In terms of hepatitis B virus data, it is noteworthy that one of the subjects exhibited a change in HBeAg from positive to negative after the 12th week of supplementation.

### 3.3. Effect of Serum 25(OH)Vitamin D Concentration on Hepatitis B Virus Data during Oligo Fucoidan Supplementation

The average serum 25(OH)Vitamin D concentration of the subjects was insufficient. To observe whether changes in the biochemical data of subjects were due to differences in their vitamin D concentrations, we divided the serum 225(OH)Vitamin D concentrations of all subjects into <20 ng/mL (Vitamin D deficiency group), 20–30 ng/mL (Vitamin D indeficiency group), and >30 ng/mL (Vitamin D normal group), and compared the relationships between their hepatitis B virus data during oligo fucoidan supplementation. The results are shown in Table 3. The results showed that in the vitamin D-adequate group, the HBV-DNA concentration of subjects receiving oligo fucoidan supplementation tended to decrease, but the number of subjects is small and the data could not reach statistical differences. There was no decreasing trend in the sufficient and vitamin D insufficiency groups. It was noteworthy that one of the subjects in the normal vitamin D group experienced a change in their HBeAg value from positive to negative after the 12th week of supplementation (Table 3).

### 3.4. Immmunocyte Markers Before andAafter Oligo Fucoidan Supplementation

During supplementation with oligo fucoidan, oligo fucoidan supplementation tended to decrease the proportion of CD4^+^, CD8^+^, CD4CD45RO^+^, and CD8CD45RO^+^ cells (Table 4). However, before supplementation with oligo fucoidan, subjects with normal vitamin D values had a higher proportion of T cells than the other two groups (Table 5). After 48 weeks of oligo fucoidan supplementation, the proportion of CD3^+^CD4^+^ and CD3^+^CD8^+^ cells increased in chronic hepatitis B patients. In particular, CD3^+^CD8^+^ cells increased in the group with normal vitamin D and the insufficient group, but in the group with sufficient vitamin D, the proportion of CD3^+^CD4^+^ and CD3^+^CD8^+^ cells were decreased. However, during the follow-up period, the proportion of CD4^+^CD45RO^+^ and CD8^+^CD45RO^+^cells tended to increase in the groups with vitamin D deficiency. All cell population of P12Wth week were higher than other groups, these results indicated that the immunomodulatory effects of oligo fucoidan could continue until the end of the supplement.

## 4. Discussion

There are some studies regarding the effect of fucoidan on the liver. Clinical studies have shown that oral fucoidan can reduce liver fibrosis and has anti-inflammatory effects in patients with non-alcoholic hepatitis (NAFLD) [17,19,21,22]. In an animal experiment, ALT and AST in the serum increased after ConA-induced liver inflammation in mice, and ALT and AST decreased in a dose-dependent manner after adding different concentrations of fucoidan [27]. Animal studies have also demonstrated that fucoidan reduces the titers of mouse liver virus B and HBsAg [28]. Fucoidan—from Cladosiphon okamuranus—is known to inhibit the replication of Hepatitis C virus (HCV)in vitro; furthermore, patients with Hepatitis C have reduced ALT and HCV RNA following fucoidan treatment [26]. This study showed that HBV-DNA decreased in patients with hepatitis B supplemented with fucoidan at 24 weeks, but did not reach statistical significance. This study also showed HBsAg had a decreasing trend with time but did not reach statistical significance. In summary, the fact that fucoidan does not specifically reduce AST and ALT may be caused by different experimental models and insufficient numbers of patients.

Fucoidan has an immune-regulating effect. Several previous studies have pointed out that fucoidan could upregulate the extracellular signal-regulated kinases (ERK) pathway in immune cells in response to diverse pro-inflammatory signals [29,30] and in hepatocytes to suppress HBV replication [28]. In an animal experiment, fucoidan enhanced immune function through the activation of natural killer (NK) cells and increased IFN-γ and IL-12 levels [31]. Studies in an elderly population in Japan have also found that 1 g of fucoidan from Undaria pinnatifida, orally administered daily, can improve the antibody responses to seasonal vaccines [32]. Other studies have shown that high levels of CD8^+^ T cells are present in the peripheral blood of patients with hepatitis B during immune tolerance [33]. In addition, studies have found that the number of CD45RO^+^ T cells in HBV patients is reduced [7,8]. The results of this study show that the CD3^+^CD4^+^ and CD3^+^CD8^+^ counts were significantly lower in patients with hepatitis B during fucoidan supplementation than before the initiation of treatment with fucoidan, but increased after treatment, and that these counts were higher than those observed before the initiation of fucoidan supplementation. We believed that fucoidan plays an immunoregulatory role in patients with hepatitis B, and the immune marker in the blood also affects the liver due to an unknown mechanism, so these markers in the blood decreased. These markers increased again without fucoidan supplementation, but further research is needed to determine the cause of this phenomenon, such as using animal experiments to observe the differences in immune markers in the liver and blood, or measuring HBV-specific T cells to solve these problems.

At present, it is believed that the main reason for the difficulty in curing chronic hepatitis B is the immunopathology of patients [10]. The immune levels of patients are closely related to nutritional status. Vitamin D can stimulate an antiviral immune response and vitamin D deficiency can increase viral replication [11]. Vitamin D deficiency is extremely prevalent and is related to the progression of liver disease [12], latitude and genetic, not seasons [34]. In this study, 88.2% of the subjects were found to be deficient or insufficient in 25(OH)Vitamin D. In the other study indicated hepatitis patients have vitamin D deficiency and follow severe bone metabolism abnormalities [35], and increased prevalence and severity of bone loss according to the severity of their liver disease [36]. In addition, fucoidan has been proven to have antiviral and immune-regulating effects in many studies. Therefore, we investigated whether the serum 25(OH)Vitamin D concentration of CHB patients would alter the effects of supplementing oligo fucoidan. It is known from these studies that the natural history of hepatitis affects the status of vitamin D, so that the season is not the most important for vitamin D concentration, so it can be speculated that the results observed after 48 weeks were caused by oligo fucoidan. In the vitamin D-adequate group, the HBV-DNA concentration of subjects receiving oligo fucoidan supplementation tended to decrease and increased the proportion of CD4^+^CD45RO^+^ and CD8^+^CD45RO^+^ cells. One of the subjects in the vitamin D-adequate group changed their HBeAg status from positive to negative after the 12th week of supplementation. The results show that vitamin D concentrations in patients with CHB seem to affect the efficacy of oligo fucoidan supplementation. When serum 25(OH)Vitamin D concentrations of patients are higher, the supplementation effect of oligo fucoidan is better.

Limitations of this study: (1) The number of subjects is small and some data cannot reach statistical differences, especially for patients with hepatitis with normal vitamin D concentrations; (2) Due to the restrictions on the admission of clinical subjects, no control group was used for oligo fucoidan treatment, and no random treatment with vitamin D was used, only the condition of patients was evaluated.

## 5. Conclusions

The above results indicate that oligo fucoidan might regulate immunity in patients with HBV infection, but the identification of the precise mechanisms underlying its immune regulatory activity requires further investigation. The vitamin D level in patients with HBV is one of the factors that might determine the effects of oligo fucoidan supplementation.

## Figures and Tables

**Table 1 nutrients-12-00321-t001:** Baseline Characteristics of the chronic hepatitis B patients.

Variable	All (*n* = 51)	
**Age**	44.86 ± 10.81	a
**BMI**	24.78 ± 4.38	a
**Gender**		b
Female	23 (45.1)	
Male	28 (54.9)	
**Education**		b
junior	5 (9.8)	
senior	16 (31.4)	
college	30 (58.8)	
**Nutrition Supplements**		b
No	19 (37.3)	
Yes	32 (62.7)	
**Exercise Per Week Frequency**		b
No	23 (45.1)	
once a week	14 (27.5)	
2–3 times a week	9 (17.6)	
over 3 times a week	5 (9.8)	
**Smoke**		b
No	48 (94.1)	
Yes	3 (5.9)	
**Drink**		b
No	38 (74.5)	
Yes	13 (25.5)	
**Vitamin D (ng/mL)**	20.6 ± 6.1	

a Data with normal distribution were presented as mean ± SD, b Categorical data were presented as *n* (%).

**Table 2 nutrients-12-00321-t002:** Between biochemistry and before and after fucoidan supplementation in chronic hepatitis B patients.

Variable	0 Week	4th Week	12th Week	24th Week	48th Week	P4Wth Week *	P12Wth Week *
Hb (g/dl)	14.0 ± 1.6	14.0 ± 1.8	14.0 ± 1.9	14.0 ± 1.7	13.8 ± 1.9	13.7 ± 1.8 ^abcd^	13.7 ± 2.0 ^bcd^
WBC (mm^3^)	5538.8 ± 1788.4	5325.0 ± 1527.9	5319.2 ± 1360.0	5247.1 ± 1589.3	5165.8 ± 1544.7 ^a^	5311.3 ± 1406.1	5582.9 ± 1525.7 ^de^
Platelets (10^3^/μL)	210.6 ± 50.7	205.3 ± 47.2	194.9 ± 49.7 ^a^	199.5 ± 51.6 ^a^	196.0 ± 45.5 ^a^	211.2 ± 52.9 ^ce^	221.7 ± 52.4 ^abcdef^
Albumin (g/dl)	4.8 ± 0.3	-	4.7 ± 0.2	4.7 ± 0.3	4.6 ± 0.3 ^ad^	4.6 ± 0.3 ^ac^	4.6 ± 0.6 ^ac^
AC sugar (mg/dl)	101.8 ± 7.6	90.1 ± 7.5 ^a^	102.5 ± 8.4 ^b^	106.4 ± 19.6 ^b^	104.7 ± 9.1 ^b^	92.5 ± 7.4 ^acde^	102.3 ± 7.1 ^bef^
ALT (IU/L)	31.3 ± 13.3	33.7 ± 26.6	32.9 ± 15.6	36.6 ± 19.1	33.0 ± 18.7	34.5 ± 18.3	34.1 ± 20.3
AST (IU/L)	25.5 ± 6.1	27.4 ± 14.8	28.9 ± 11.5	31.7 ± 14.6 ^a^	31.9 ± 32.5	25.8 ± 5.9	27.2 ± 10.9
TG (mg/dl)	92.0 ± 58.9	99.2 ± 43.9	97.0 ± 56.9	88.8 ± 54.6	92.2 ± 55.1	100.0 ± 45.8 ^d^	88.4 ± 48.4
Cholesterol (mg/dl)	190.7 ± 39.9	183.0 ± 41.8	194.6 ± 45.2	186.3 ± 43.3	182.3 ± 37.0 ^c^	178.3 ± 34.6 ^ac^	182.3 ± 36.1
Creatinine (mg/dl)	0.78 ± 0.18	0.82 ± 0.20 ^a^	0.76 ± 0.18 ^b^	0.80 ± 0.20 ^c^	0.78 ± 0.18 ^b^	0.84 ± 0.18 ^ace^	0.79 ± 0.19 ^f^
HBV-DNA (log 10 IU/mL)	3.5 ± 1.6	-	3.5 ± 1.8	3.3 ± 1.7	3.6 ± 1.5	-	3.6 ± 1.6
HBsAg (IU/mL)	1817.7 ± 3974.6	-	2178.9 ± 5234.3	2417.2 ± 6648.8	1676.1 ± 3424.6	-	1792.6 ± 3936.9
HBeAg positive (n, %)	4/51 (7.8)	-	3/51 (5.9)	3/51 (5.9)	3/45 (6.7)	-	2/26 (7.7)

Data were continuous measurement and presented as mean ± SD using repeated measure of general linear model. Hb: Hemoglobin; WBC: White blood cell; AC sugar: Ante Cibum (before meals) sugar; ALT: Alanine aminotransferase; AST: Aspartate aminotransferase; TG; Triglyceride. ^a^
*p*-value < 0.05: Compared with 0 week; ^b^
*p*-value < 0.05: Compared with 4th week; ^c^
*p*-value < 0.05: Compared with 12th week; ^d^
*p*-value < 0.05: Compared with 24th week; ^e^
*p*-value < 0.05: Compared with 48th week; ^f^
*p*-value < 0.05: Compared with P4Wth week. * P4Wth: 4 weeks after the supplementation; P12Wth: 12 weeks after the supplementation.

**Table 3 nutrients-12-00321-t003:** Effect of serum 25(OH)Vitamin D concentration on hepatitis B virus data during oligo fucoidan supplementation.

Variable	0 Week	4th Week	12th Week	24th Week	48th Week	P4Wth Week	P12Wth Week
Vitamin D normal group							
HBV-DNA (log 10 IU/mL)	5	-	4.7	4.3	4.3	-	4
HBsAg (IU/mL)	272.8 ± 18.1	-	394.4 ± 25.0	297.8 ± 65.8	282.1 ± 30.9	-	266.9 ± 36.8
HBeAg postive (n, %)	1/6 (16.7)	-	0/6 (0)	0/6 (0)	0/4 (0)	-	0/2 (0)
Vitamin D indeficiency group							
HBV-DNA (log 10 IU/mL)	3.8 ± 1.8	-	3.8 ± 2.1	3.6 ± 2.0	3.9 ± 1.8	-	3.9 ± 2.0
HBsAg (IU/mL)	2515.3 ± 5240.7	-	3114.7 ± 6938.3	3497.3 ± 8845.9	2281.9 ± 4501.3	-	2453.4 ± 5186.8
HBeAg postive (n, %)	1/20 (5)	-	1/20 (5)	1/20 (5)	1/19 (5.3)	-	1/14 (7)
Vitamin D deficiency group							
HBV-DNA (log 10 IU/mL)	3.0 ± 1.3	-	3.0 ± 1.1	2.8 ± 1.0	3.1 ± 1.0	-	3.1 ± 1.0
HBsAg (IU/mL)	1083.8 ± 1090.1	-	1142.3 ± 1133.4	1231.5± 1446.4	1048.2 ± 1017.8	-	1108.8 ± 1168.3
HBeAg postive (n, %)	2/25 (8)	-	2/25 (8)	2/25 (8)	2/22 (9)	-	2/10 (20)

Data were continuous measurement and presented as mean ± SD using repeated measure of general linear model. P4Wth: 4 weeks after the supplementation; P12Wth: 12 weeks after the supplementation.

**Table 4 nutrients-12-00321-t004:** Between CD (Cluster of differentiation) marker and before and after fucoidan supplementation in chronic hepatitis B patients.

Variable (%)	0 Week	4th Week	12th Week	24th Week	48th Week	P4Wth Week	P12Wth Week
CD3^+^CD4^+^	10.0 ± 6.3	9.4 ± 7.7	7.7 ± 4.3	7.5 ± 4.8	9.5 ± 5.9	9.7 ± 3.5 ^cd^	11.7 ± 6.2 ^cd^
CD3^+^CD8^+^	8.1 ± 6.1	8.1 ± 5.6	5.8 ± 4.3 ^a^	6.7 ± 5.8	9.4 ± 5.8 ^cd^	9.5 ± 6.1 ^cd^	10.1 ± 6.7 ^cd^
CD4^+^ CD45RO^+^	5.2 ± 6.7	4.9 ± 2.9	5.0 ± 2.2	4.7 ± 2.8	4.2 ± 2.5	4.8 ± 3.4	5.9 ± 5.7
CD8^+^CD45RO^+^	2.2 ± 4.7	1.3 ± 1.0	1.8 ± 1.8	2.0 ± 1.3 ^b^	2.1 ± 2.8	1.4 ± 1.3	3.4 ± 4.2 ^bf^

Data were continuous measurement and presented as mean ± SD using repeated measure of general linear model. ^a^
*p*-value < 0.05: Compared with 0 week. ^b^
*p*-value < 0.05: Compared with 4th week. ^c^
*p*-value < 0.05: Compared with 12th week. ^d^ p-value < 0.05: Compared with 24th week. ^e^
*p*-value < 0.05: Compared with 48th week. ^f^
*p*-value < 0.05: Compared with P4Wth week. P4Wth: 4 weeks after the supplementation; P12Wth: 12 weeks after the supplementation.

**Table 5 nutrients-12-00321-t005:** Effect of serum vitamin D concentration on CD marker data during oligo fucoidan supplementation.

Variable (%)	0 Week	48th Week	P4Wth Week	P12Wth Week
Vitamin D normal group				
CD3^+^CD4^+^	16.5 ± 0.1	10.6 ± 0.4 ^a^	9.6 ± 1.3	11.2 ± 1.5
CD3^+^CD8^+^	10.6 ± 13.6	14.3 ± 7.5	6.9 ± 0.1	16.8 ± 1.3
CD4^+^ CD45RO^+^	12.3 ± 9.8	4.9 ± 3.5	2.2 ± 0.9 ^c^	8.4 ± 4.4
CD8^+^CD45RO^+^	10.9 ± 15.3	5.9 ± 7.1	1.2 ± 0.0	6.8 ± 2.9
Vitamin D indeficiency group				
CD3^+^CD4^+^	8.7 ± 5.5	11.4 ± 5.8	10.6 ± 2.8	13.0 ± 6.0 ^ac^
CD3^+^CD8^+^	7.0 ± 5.4	10.9 ± 6.0 ^c^	6.9 ± 0.1 ^ac^	9.9 ± 5.5 ^c^
CD4^+^ CD45RO^+^	5.3 ± 7.9	4.7 ± 2.1	5.6 ± 3.9	5.6 ± 5.2
CD8^+^CD45RO^+^	1.6 ± 3.1	2.2 ± 2.7	1.2 ± 1.1	3.4 ± 4.2
Vitamin D deficiency group				
CD3^+^CD4^+^	10.8 ± 7.3	6.6 ± 5.4	8.4 ± 4.4 ^d^	9.8 ± 6.8 ^d^
CD3^+^CD8^+^	9.2 ± 6.2	6.3 ± 4.0	8.9 ± 6.2 ^d^	9.3 ± 8.4 ^d^
CD4^+^ CD45RO^+^	3.6 ± 3.2	3.4 ± 2.9	4.4 ± 2.8	5.7 ± 6.9
CD8^+^CD45RO^+^	1.3 ± 0.9	1.4 ± 1.1	1.7 ± 1.7	2.9 ± 4.4

Data were continuous measurement and presented as mean ± SD using repeated measure of general linear model. ^a^
*p*-value < 0.05: Compared with 0 week; ^b^
*p*-value < 0.05: Compared with 4th week; ^c^
*p*-value < 0.05: Compared with 12th week; ^d^
*p*-value < 0.05: Compared with 24th week.

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
