# Peer review of "The 25(OH)Vitamin D Status Affected the Effectiveness of Oligo Fucoidan in Patients with Chronic Hepatitis B Virus Infection with Immune Tolerance Phase"

_nutrients, 2020, doi:10.3390/nu12020321_

Round 1
Reviewer 1 Report
This is a clever study that seeks to identify the role of vitamin D status in oligo-fucoidan treatment of hepatitis B. The obvious deficiencies of the study are that there was no control group for the oligo-fucoidan treatment and no randomized vitamin D treatment. Apart from that, this study should be thought of as an interesting pilot study of this effect, now worthy of more serious study.
Major Concerns
The two independent variables are the oligo-fucoidan treatment and the vitamin D status of the hepatitis patients. The weakness here is that there was no control group for the oligo-fucoidan treatment and no randomized treatment with vitamin D, only an assessment of patient status. But the question then remains, can we learn anything important and interesting from this study, which is essentially a pilot study on this issue. I think the answer is yes, if these issues are made clear and the results properly expressed.
The study is weakly powered to examine the interaction of the two independent variables, which makes it difficult to state definitively the effects of either treatment or the interaction. The secular trend in the natural history of hepatitis and the seasonal trend in vitamin D status makes it difficult to be persuaded that the results that were seen after 48 weeks were due to the oligo-fucoidan or vitamin D.
The authors do not spend much time explaining any mechanisms for the oligo-fucoidan, a sulfated polysaccharide. Is it even absorbable? In what form? How does it work?
Suggestions:
A paragraph is needed at the end of the Discussion in which the authors acknowledge clearly the design deficiencies of this study.
The authors need to include much more in the Methods section, starting with the method of vitamin D analysis and the splitting of the cohort by vitamin D status. The methods for almost all of the variables in Table 2 are not described in the Methods. The authors describe the analysis of cytokine levels but then show no results of these analyses. This section can be removed.
The tables are a mess. I suggest that Tables 3 and 4 should immediately follow Table 2 in the text. Section 3.3 describes the separation of the group by vitamin D status and then refers to Table 3, which is not divided by vitamin D status. Lines 154 is probably referring to Table 4, not Table 3. But then the sequence of tables in the text is out of order. The first line of the next section, line 161-162 is, in fact, referring to Table 3, not Table 4.
Many times the results are overstated. In lines 154-156 we find the following, “The results are shown in Table 3. The results showed that in the vitamin D adequate group, the HBV-DNA concentration of subjects receiving oligo fucoidan supplementation tended to decrease.” With no SD and no p-values, this is more hope than fact, especially considering the significant secular trend probably at work here. The authors need to be much more circumspect of the interpretation of their results here and elsewhere when the p-value is not significant.
Minor Concerns
Line 17: “investigated the 25(OH)Vitamin D” should probably be “investigated how 25(OH)Vitamin D”
Line 21 and elsewhere especially Table 2: please define your abbreviations such as “WBC, PLT”. Also GPT, AC, GOT, TG, Hb.
Line 71: Fucoidan is a sulfated polysaccharide
Line 90: ALT was used as an inclusion criteria. It would be nice to see these results for the cohort in Table 1.
In section 2.5. Statistical analysis, considerable time was spent describing the nonparametric testing that was used. However, the only statistics that were used in Tables 2-5 were Repeated measure of General linear model. The authors can remove most of the text in that section.
Line 129: What does it mean that “the presence of HBsAg was greater than 6 months?” Maybe longer than 6 months?
Line 206-207: Please rephrase. On first reading this, it appeared to me that the authors were referring to the subjects of this study as “These hepatitis patients”. It took a look at the reference list to realize that they were referring to patients from another continent.
Author Response
Reply to reviewer1's comments
Major comments
Limitations on research have been added in the discussion section: Limitations of this study (1) The number of subjects is small and some data cannot reach statistical differences, especially for patients with hepatitis with normal vitamin D concentrations. (2) Because of the restrictions on the admission of clinical subjects, there is no control group was used for oligo fucoidan treatment, and no random treatment with vitamin D was used, just only the condition of patients was evaluated. (p12) Vitamin D deficiency is extremely prevalent and is related to the progression of liver disease in chronic liver disease [12], latitude and genetic, not seasons [35]. It is known from these studies that the natural history of hepatitis affects the status of vitamin D, so that the season is not the most important for vitamin D concentration, so it can be speculated that the results observed after 48 weeks were caused by oligo fucoidan. (p11) Fucoidan has an immune-regulating effect. Several previous studies have pointed out that fucoidan could upregulate ERK pathway in immune cells in response to diverse pro-inflammatory signals [29,30] and in hepatocytes to suppress HBV replication [31]. (p11)
Minor comments
“Fucoidan is a polysaccharide” has been corrected to “Fucoidan is a sulfated polysaccharide” The level of 25(OH)Vitamin D was measured by Chemiluminescence assay. The serum 25(OH)Vitamin D concentrations of all subjects divided into <20 ng/ml (Vitamin D deficiency group), 20-30 ng/ml (Vitamin D indeficiency group), and >30 ng/ml (Vitamin D normal group) We deleted “Analysis of cytokine levels” and corrected “Statistical analysis” “the presence of HBsAg was greater than 6 months” has been corrected to “the presence of HBsAg was sustained than 6 months” We reordered the position of the table Hb: Hemoglobin; WBC: White blood cell; AC sugar: Ante Cibum(before meals) sugar; ALT : Alanine aminotransferase ; AST : Aspartate aminotransferase; TG ; Triglyceride were added in Table 2.
All other replies were corrected in the manuscript. Thanks a lot.

Reviewer 2 Report
In ko’s study, fifty-one patients with HBV infection received oligo fucoidan (550 mg) twice a day for 48 weeks. The results indicated that oligo-fucoidan may regulate immune effects in patients with HBV infection, and the 25(OH)Vitamin D status affected the effectiveness of oligo fucoidan. Some comments were shown below:
1. line 88: typo is needed to be corrected.
2. Material and method: How do you detect vitamin D levels in patients?
3. line 107: anti-human CD45RO was repeated.
4. Table 1: Did patients take any medicine for HBV infection? What kind of nutrition supplements were taken for 32 subjects? Could nutrition supplements possible affect the results?
5. Table 2: the statistical results of P4Wth week showed Hb: 13.7 ± 1.8abcd, Platelets 211.2 ± 52.9ce, AC sugar 92.5 ± 7.4acde, TG 100.0 ± 45.8d, Cholesterol 178.3 ± 34.6ac, Creatinine 0.84 ± 0.18ace. It is clear the rational and significance of compared data of P4Wth week with the data of 12th, 24th, and 48th weeks. Some parameters were higher than the results of 12th, 24th, and 48th weeks, but some are decreased. What would you like to illustrate from the results?
7. Table 3: How do you define Vit D concentration of normal, indeficiency, and deficiency?
8. line 158 and 159, this maybe a wrong label (could be table 3 not table 4).
9. Table 4: CD4+ and CD8+ did were not decreased at 48th week. How do you explain this? Moreover, all cell population of P12Wth week were higher than other groups.
Author Response
Reply to reviewer2's comments
How to measure the level of 25(OH)Vitamin D: The level of 25(OH)Vitamin D was measured by Chemiluminescence assay. The serum 25(OH)Vitamin D concentrations of all subjects divided into <20 ng/ml (Vitamin D deficiency group), 20-30 ng/ml (Vitamin D indeficiency group), and >30 ng/ml (Vitamin D normal group). (p3) About 62.7% of patients using Nutrition supplements: These patients take any medicine for HBV infection. 62.7% of patients were supplemented with nutritional supplements, of which 85% were supplemented with comprehensive vitamins and minerals. (p3) During oligo fucoidan supplementation, the levels of WBC, Hb and Platelets more decreased than baseline, but these data were in the normal range. AC sugar, TG and Cholesterol did not show an upward or downward trend during the supplementation. (p4) All cell population of P12Wth week were higher than other groups, these results indicated that the immunomodulatory effects of oligo fucoidan could continue until the end of the supplement.(p9)
All other replies were corrected in the manuscript. Thanks a lot.

Round 2
Reviewer 1 Report
A number of significant improvements have been made in this interesting manuscript. There are several small changes that should be included. The conclusions, both in the Abstract and in the Conclusion still overstate the certainty the data allow in the effect of vitamin D on the oligo fucoidan effectiveness.
Abstract
Line 27, Change “affected” to “might have affected”
Methods
The standard of scientific research is that studies are reported in enough detail so as to be replicable. The methods for the analytes in Table 2 are largely not described in the methods. More than likely, these were performed in a clinical lab by an automated system. Sufficient for the purposes of this paper would be to describe the analytical system used in the lab. The method used to quantify HBV DNA the concentrations of HBsAg, anti-HBs, HBeAg, and anti-HBeAg were also not described.
Results
The text in lines 124-135 is confusing. Lines 124-131 describe the results in Table 1 as indicated. But then the text in lines 131-134 describe results from Table 2, but this is not indicated. Then, the last sentence in lines 134-135 go back to Table 1 results. This must be corrected.
In the text, it appears that lines 137-147 are bolded. The following sentence, “Blood was collected from the 51 subjects to measure biochemical values at week 0 (before the experiment), and the 4th, 12th, 24th, 48th weeks (in the experiment) and in the 4th and 12th weeks after the experiment.” described the methods used and should be moved there. The following sentence, “During oligo fucoidan supplementation, the levels of WBC, Hb and Platelets more decreased than baseline, but these data were in the normal range.” would be better and more accurately written, “During oligo fucoidan supplementation, the levels of WBC, albumin and Platelets decreased from baseline, but these data were in the normal range.”
Discussion
In line 229, start “Limitations…” on a new paragraph.
Line 232, Remove “there is”
Line 233, Remove “just”
Conclusion
Line 238, Change “determines” to “might determine”
Author Response
Reply to reviewer1's comments
Abstract: We changed “affected” to “might have affected” Methods: Serum HBV DNA levels (copies/mL) were measured by polymerase chain reaction (PCR) using Cobas TaqMan HBV Test, v2.0 (Roche Molecular Systems, CA, USA), according to the manufacturer’s instructions. Quantification of HBsAg was performed by an automated chemiluminescent micro-particle immunoassay (CMIA) using the Roche Cobas e602 analyzer with Elecsys HBsAg II Quant reagent kits (Roche Diagnostics, CA, USA), according to the manufacturer’s instructions. Quantification of HBeAg was conducted by an automated chemiluminescent micro-particle immunoassay (CMIA) using the Roche Cobas e602 analyzer with Elecsys HBeAg Quant reagent kits (Roche Diagnostics, CA, USA), according to the manufacturer’s instructions (p3). Results: We corrected the mistakes. With respect to 25(OH)VitaminD, the average serum 25(OH)Vitamin D concentration in the patients was 20.6 ± 6.1 ng/ml, which is inadequate (Table 1). Higher levels of WBC, PLT, and albumin were recorded at the baseline than after 48 weeks of fucoidan administration (p <0.05). Higher levels of AC sugar were recorded at the baseline than after 4 weeks of fucoidan administration (p <0.05) (Table 2). In the text: We corrected the mistakes (p3 and p4). In discussion and conclusion: We corrected the mistakes (p11).

Reviewer 2 Report
I have no other comments.
Author Response
Thank you very much!